# Exploring General Practitioners’ Views and Experiences of Providing Care to People with Borderline Personality Disorder in Primary Care: A Qualitative Study in Australia

**DOI:** 10.3390/ijerph15122763

**Published:** 2018-12-06

**Authors:** Julian Wlodarczyk, Sharon Lawn, Kathryn Powell, Gregory B. Crawford, Janne McMahon, Judy Burke, Lyn Woodforde, Martha Kent, Cate Howell, John Litt

**Affiliations:** 1College of Medicine and Public Health, Flinders University, GPO Box 2100, Adelaide 5001, Australia; wlod0002@flinders.edu.au; 2Flinders Human Behaviour and Health Research Unit, Flinders University, GPO Box 2100, Adelaide 5001, Australia; 3Faculty of Health and Medical Sciences, University of Adelaide, Adelaide 5000, Australia; powkj001@gmail.com; 4North Adelaide Palliative Care Service, Discipline of Medicine, University of Adelaide, Adelaide 5000, Australia; gregory.crawford@adelaide.edu.au; 5Private Mental Health Consumer Carer Network (Australia) Ltd., PO Box 542, Marden 5070, Australia; jmcmahon@senet.com.au; 6Sanctuary BPD Carer Support, Adelaide 5001, Australia; bobandjudy@adam.com.au; 7Carers SA, 338 Tapleys Hill Rd, Seaton 5023, Australia; colyn@internode.on.net; 8Borderline Personality Disorder Centre of Excellence, Country Health SA Mental Health Services, 22 King William St, Adelaide 5000, Australia; martha.kent@sa.gov.au; 9Cate Howell, Cate Howell and Colleagues, 14 Hay St, Goolwa 5214, Australia; cate.howell@gmail.com; 10Department of General Practice, College of Medicine and Public Health, Flinders University, GPO Box 2100, Adelaide 5001, Australia; jlitt@flinders.edu.au

**Keywords:** borderline personality disorder, primary care, general practitioners, mental health services, mental illness, qualitative research

## Abstract

The prevalence of people seeking care for Borderline Personality Disorder (BPD) in primary care is four to five times higher than in the general population. Therefore, general practitioners (GPs) are important sources of assessment, diagnosis, treatment, and care for these patients, as well as important providers of early intervention and long-term management for mental health and associated comorbidities. A thematic analysis of two focus groups with 12 GPs in South Australia (in discussion with 10 academic, clinical, and lived experience stakeholders) highlighted many challenges faced by GPs providing care to patients with BPD. Major themes were: (1) Challenges Surrounding Diagnosis of BPD; (2) Comorbidities and Clinical Complexity; (3) Difficulties with Patient Behaviour and the GP–Patient Relationship; and (4) Finding and Navigating Systems for Support. Health service pathways for this high-risk/high-need patient group are dependent on the quality of care that GPs provide, which is dependent on GPs’ capacity to identify and understand BPD. GPs also need to be supported sufficiently in order to develop the skills that are necessary to provide effective care for BPD patients. Systemic barriers and healthcare policy, to the extent that they dictate the organisation of primary care, are prominent structural factors obstructing GPs’ attempts to address multiple comorbidities for patients with BPD. Several strategies are suggested to support GPs supporting patients with BPD.

## 1. Introduction

People with Borderline Personality Disorder (BPD) have some of the highest levels of unmet need in mental health services [1]. BPD is associated with greater demands for consultations, telephone calls, and requests for medications in primary health care for family physicians or general practitioners (GPs) [2,3]. The estimated 12-month prevalence of BPD in the general Australian population of 2%, or over 440,000 Australians, is believed to be a gross underestimation [4]. Lifetime prevalence is estimated to be up to 6% [5]. The prevalence in primary care has been assessed to be four to five times higher than in the general population [3,6]. Researching BPD within primary care is important, given that “a great deal of mental distress, especially that of a chronic nature, either never reaches secondary care or, if it does, individuals will usually have intermittent contact with psychiatrists” (p. 91) [7].

BPD is under-recognised in primary care [4,8]. The diagnosis of BPD is a significant challenge for GPs; because of the nature of the condition, patients frequently have comorbid conditions that may take precedence, and BPD is an extremely challenging condition psychologically for GPs [3]. The comorbid conditions can be both mental and physical, creating a complex situation when there is already a well-acknowledged shortage of resources, including specialist psychiatric services [9].

The Australian National Health and Medical Research Council (NHMRC) Clinical Practice Guideline for the treatment of BPD, which is aligned with the international evidence base, emphasises the importance of GPs in the diagnosis and development of management plans for these patients [2,3,4,10,11]. Since people with BPD may also suffer from many chronic medical disorders as well as high rates of mental illness, the role of the GP is vital in assessing and coordinating care [2,12,13]. The NHMRC Guideline identifies three major deficiencies in the Australian health service system for BPD: that it is under-recognised within primary health care sector and that health professionals are often not aware of [locally] available services for BPD; that there is a need to develop a crisis plan for the BPD patient, as well as the families and carers of people with BPD, in order to assist them to navigate existing services; and thirdly, these gaps in need result in a high presentation at emergency departments. 

GPs often find that clinical relationships with these patients are challenging, because the behaviours arising from their BPD can be extremely counterproductive for care management [2,3]. People with BPD may suffer significant mood swings, have great interpersonal difficulties, and demonstrate self-harm and suicidal behaviours [2,4,12]. A United Kingdom (UK) study of GPs’ views and experiences about suicide risk assessment and the management of young people in primary care found that GPs believed that they lacked specialist knowledge and clinical skills. They also believed that suicide was “very difficult to predict and therefore unpreventable, which left them feeling powerless to influence the course of events” [14]. Clarifying these challenges is necessary to improve healthcare outcomes in the primary care setting, and improve GPs’ confidence and capacity to provide care and support to people with a BPD diagnosis.

Despite the high and consistent use of health care services, BPD often goes unrecognised. In a United States (US) study, up to 42.9% of patients with BPD were not identified by their primary care physician (i.e., GP) as having continuing emotional or mental health issues [6]. Whilst many patients with BPD describe the positive aspects of the care that they have received from GPs [12,15], multiple studies over more than four decades have confirmed that people with BPD are often stigmatised by employees at all levels of healthcare systems [16]. An Australian study with a convenience sample of consumers (*n* = 413) and carers (*n* = 200) found that over half of those surveyed reported feeling as though they had been “shunned” by health professionals. Furthermore, 28.3% of mental health consumers felt that they experienced “bad” care from GPs [17]. A more recent study comparing health professionals [18] found that GPs were more likely than other professionals to hold greater beliefs about the person’s dangerousness and personal weakness. These figures illustrate the likelihood of there being an association between people with BPD experiencing difficulties in receiving satisfactory and respectful care from the health system, and the challenges experienced by health professionals when treating patients with BPD. 

One of the most important aspects of care for people with BPD is the need for continuity of care and consistent support. GPs have been clearly identified as important in providing such consistency [12]. However, more understanding of the needs of GPs to help them overcome the challenges in providing care to BPD patients is needed. Improving GPs’ provision of evidence-based care for BPD patients has clear financial value to the healthcare system [19], and GPs are well-placed to help improve the quality of life for this significant group of patients.

## 2. Materials and Methods

This study was designed to explore the nature and difficulties for GPs, examine the reasons that caring for people with BPD in primary care is so difficult and not well managed, generally. It also explores what strategies and actions might assist with improving the care of their patients with BPD. Two focus group discussions were coordinated in April 2016 in central Adelaide at a recognised GP support and training venue (GP Partners Australia). 

In Australia, general practice settings and mental health services are quite distinct service types that are funded differently: general practices are privately-run businesses with a limited though growing colocation of other disciplines and service types; and mental health services are mostly government-funded and multidisciplinary, with some dedicated private psychiatric inpatient care, and private psychiatrists and psychologists located sporadically in the community. Communication and referral pathways can be fragmented and difficult to navigate between these service types, and electronic shared mental health records do not currently exist between general practice and these mental health services.

Participation in the current study was open to any currently practicing GP. The two-hour focus groups were held one week apart to offer flexibility in attendance, and were preceded with a light meal, given that the GPs were likely arriving directly following their working day. Participants included 12 GPs and the research team: a collaboration of stakeholders that comprised three university-based researchers with an interest in BPD, a psychiatrist specialising in BPD treatment and care, an academic physician, a GP academic also involved in delivering Royal Australian College of General Practice (RACGP) national education, two people with BPD, and three family carers (total *n* = 10). This collaborative mix of stakeholders came together as a consequence of their shared interest and advocacy concerning BPD care. They formed a dedicated ‘research’ team, meeting regularly over several weeks to plan and co-design this research activity, including the design of the focus group sessions with GPs, the questions asked of the focus groups, and co-facilitation of the event. Their presence as a team was to enhance the focus group discussions and demonstrate their shared commitment to understanding the issues being explored. This deliberate design of the focus groups aligned with GPs’ expectations for learning events to be informative, educational, and provide the opportunity for peer learning and sharing, as well as enabling discussion and reflection within an environment that enabled direct access to experts in the field of focus [20]. GPs were recruited through GP Partners Australia’s monthly electronic newsletter to approximately 1500 GP members. 

The discussions were audio-recorded and centred on the challenges of addressing BPD in the primary care setting. A moderator [KP] guided discussions using guide questions developed collaboratively by the research team. The focus group questions are outlined in Table 1. All of the members of the research team were encouraged to take field notes to enable a robust group discussion of focus group data during the analysis stage.

The focus group discussions were professionally transcribed, and the resulting anonymised text coded to identify repeating ideas using Template Analysis [21]. This involved text analysis initiated according to certain a priori codes believed by the main researchers to bear on the analysis (e.g., ‘*doctor–patient relationship’, ‘systemic shortcomings’, ‘familiarity with disorder’*). Some of these codes were modified, and others were created (e.g., ‘*comorbidity*’) as a response to prevalent topics identified during analysis. Hierarchical relationships were established, allowing groups of codes to be subsumed under others, and certain codes were collapsed into each other following discussion. 

Once preliminary codes were determined, two members of the research team [SL and JW] met to discuss, debate, and formulate a mind map to represent the potential overarching themes and sub-themes. This visual process was chosen because it enabled the researchers to determine the range of themes and make clear analysis decisions about the order of themes and sub-themes, as well as the linkages between them, in order to make sense of the complexity of issues investigated [22]. This mind map was then presented to the larger research group for further discussion and refinement prior to finalising the analysis. By involving all of the members in this way, quality checking was carried out to maintain reflexivity and avoid bias. Given the focus of the study was on GPs’ experiences, only GP data was analysed and reported.

The research was approved by the University of Adelaide Human Research Ethics Committee (No. H-2015-022). Each participant was provided with a participant information sheet, and provided written consent for participation at the beginning of the focus group.

## 3. Results

Twelve GPs participated in this study, including an equal mix of female and male GPs. Specific demographic information was not collected; however, the sample included GPs in solo, small, and larger practices, in inner city and outer suburban locations, and with varying lengths of practice experience as a GP.

Four major themes arose from the data analysis. These were: (1) Challenges Surrounding Diagnosis of BPD; (2) Comorbidities and Clinical Complexity; (3) Difficulties with Patient Behaviour and the GP–Patient Relationship; and (4) Finding and Navigating Systems for Support. The themes were linked by an apparent lack of resources and BPD-specific skills among GPs, despite the willingness that they expressed to improve their confidence and abilities regarding this patient group. See the mind map in Figure 1 for a graphical representation of the issues and their relationships. Direct quotes from the GP participants are used to demonstrate each theme, where FG1 and FG2 refer to each focus group discussion. GP demographic information was not collected.

### 3.1. Challenges Surrounding Diagnosis of BPD

GPs repeatedly identified and highlighted that diagnosing BPD as a discrete psychiatric entity was obstructed by the pressures experienced during multi-morbid, complex presentations. In cases involving undiagnosed patients, GPs found themselves attending to a range of symptomatology in what appeared to be time-limited conditions (reported sporadically by the patient), prohibiting a clear diagnosis. Some GPs recognised this as a barrier because they claimed that establishing a diagnosis is the first step in determining an appropriate management plan.
“They come in with some sort of behavioural problem, or depression, or anxiety, or a mood swing, and then you’ve got to try and work out whether they might fit into a broader category.”(GP, FG2)

In other cases, the possibility that a personality disorder was at play stood out more to some GPs by virtue of clinical experience and greater familiarity with some of the core features that characterise presentations involving a BPD diagnosis. GPs talked about how a lack of knowledge and skills contributed to diagnostic problems. Within the group, there was a range of confidence and skill levels, which made them uncertain about making the BPD diagnosis.
“…the things that alert me are some of the textbook criteria of people who talk about a lot of empty feelings, a lot of “what if somebody abandons me”, the rapid change—one minute thinking someone’s wonderful and the next minute saying how awful they are, which sometimes I discover applies to me as well.”(GP, FG1)
“But it’s complicated too, because not all ‘cutters’ are necessarily BPD. So, it’s not straightforward…”(GP, FG1)

GPs spoke of their reliance on clinical “instinct” in diagnosing BPD. In lieu of a formal diagnostic approach, they used “clues” emerging from the patient’s presentation during GP–consumer interactions, and the overall “gut feeling” that something else may be operating in the individual’s clinical picture. With increasing visits by the patient, the GP would start to see a pattern and suspect some underlying problem.
[When asked how they recognize BPD] “You get sort of an instinct for it and the nature of the interaction I think gives you the clues…and I guess you get that gut feeling.”(GP, FG1)

On the other hand, some GPs claimed that their clinical judgment or instinct failed to detect the existence of an underlying personality disorder. They found themselves completely engaged with the several aspects of an individual’s presentation, such that questions around diagnosing BPD and applying evidence-based care premised on a diagnosis of BPD did not surface during consultation. These remarks related to several comorbidities presenting in a short amount of time.
[On encountering a complex presentation possibly involving BPD] “… I’m seeing the presenting things and not actually looking underneath.”(GP, FG1)
“I feel so bogged down sometimes with the physical symptoms that they’ve presented with that I never feel like I get to the psychological.”(GP, FG1)

GPs noted that many of their first encounters with individuals with BPD occurred during a patient crisis. The urgent and emergency-laden nature of the GP–patient interaction that took place in these circumstances precluded a psychiatric evaluation that would otherwise enable a BPD diagnosis. In their ongoing care of these patients, GPs also described frequently providing care for the patients’ immediate needs arising out of crisis, rather than unpacking a possible personality disposition that provokes a crisis in the first place. Therefore, GPs were not in a position to properly diagnose BPD because of the common presentation circumstances of this patient group.
“… meeting these people for the first time, you’re usually dealing with the crisis that they come in with rather than the diagnosis.”(GP, FG2)

GPs identified problems around diagnosis in multi-morbid presentations involving individuals with BPD as particularly relevant for more junior GPs still in their early clinical careers. GPs said that a lack of experience impedes prioritisation and leads to delays in diagnosis.
“I think the difference with less experience is that the physical things that present are so front and central that they [junior doctors] find it difficult to even look to something else before they’ve gone through each and every one of the physical things…”(GP, FG1)

Furthermore, some participants explained that, even in cases where a BPD diagnosis was previously made by another healthcare provider, the GP role was not necessarily made any more straightforward: it was pointed out that some patients avoid disclosing a history of BPD, leading to delays in or a lack of diagnosis. GPs tied this problem to the well-known stigma that is associated with BPD in healthcare circles and elsewhere.
“I reckon a lot of people sort of deny it… Not that many patients come in and say they’ve got a personality disorder, because I think there is a bit of a stigma there.”(GP, FG2)

One GP commented that a comprehensive evidence-based clinical practice guideline is necessary for diagnosis and management, given that practice guidelines are available for most other medical conditions. This comment was made in spite of the existence of the 2012 NHMRC Clinical Practice Guideline for BPD.
“We must have a consensus statement on what the evidence-based interventions are that you can use in these conditions. Most diseases have a manual plan, but it doesn’t seem to be working for many psychological and psychiatric conditions. Yeah, it would be very difficult to do, but at least something that says ‘these five therapies work’.”(GP, FG2)

### 3.2. Comorbidities and Clinical Complexity

The issue of several presenting complaints resulting from co-occurring chronic conditions was frequently discussed. GPs unanimously felt that presentations involving a diagnosis of BPD are clinically overwhelming due in part to the high rates of comorbidity that this patient group experiences. GPs repeatedly mentioned several other mental health diagnoses that they frequently encounter among individuals with BPD, such as mood disorders, anxiety disorders, substance use disorders, trauma and stress-related disorders, and eating disorders. Importantly, they emphasised that individuals with BPD frequently also present with several medical complaints in addition to mental health complaints, ranging from abdominal discomfort to chest pain. Whilst having to address both the medical and mental health aspects of a patient’s presentation was described as a ‘balancing act’ that GPs were familiar with in their role, they described the level of complexity for BPD patients as particularly stressful and prone to inadequate primary care management.
“Comorbidity seems to be huge. I mean they present with so many physical symptoms as well as mental health symptoms... it’s not unusual to have six, seven, eight, nine, separate problems, and the last one will be, ‘oh, I’ve been having chest pains’.”(GP, FG1)
[Of treating comorbidities] “… you’re committed to investigating and I try to prioritise them, what the potential life-threatening things are, and then work my way through it.”(GP, FG2)

GPs identified the combination of medical and mental health problems as a significant management barrier, given the time constraints of practice. There was a clear consensus that the amount of time afforded during standard consultations is insufficient to appropriately address the number of complaints with which individuals BPD present. Even in cases where longer consultations are arranged in anticipation of such complexity, GPs often thought that the amount of time allotted falls short of what is required to effectively prioritise and confront multifaceted presentations. Healthcare policy—to the extent that it dictates the temporal organisation of primary care—clearly emerged as a prominent structural factor obstructing GPs’ attempts to properly address multiple comorbidities.
“I’ve made an hour-long appointment for a patient because she has so many problems; then she didn’t turn up.”(GP, FG1)
“Why the short appointments? Why is Medicare loaded towards the short appointments?”(GP, FG2)
[Of time constraints while addressing multiple issues] “… can we just clarify, the therapy has 15 minutes to gauge this, and you probably have half an hour to 45 minutes just to sort of make sure you’re in the right playing field.”(GP, FG1)

Problems involving the prioritisation of complaints are worsened by what one GP called the “*compelling*” nature of the issues with which individuals with BPD often present. Participants referred to a sense of intensity and urgency regarding whichever complaint(s) that the individual emphasised, which may have been at odds with the GPs’ perspectives, and clinical instincts, and which therefore may have distracted them from focussing on longer-term management goals. GPs also expressed the view that adding another diagnosis simply may detract from a potential BPD diagnosis. GPs expressed uncertainty as to what should be addressed during each consultation. They described feelings of being overwhelmed and powerless to effect long-term improvement in this patient group.
[Concerning the nature of patient presentations] “… the way they present with physical symptoms can be very compelling… Someone’s feeling whatever symptoms they’re presenting with can be very engaging for us.”(GP, FG1)
[Concerning mental health comorbidities] “… there’s often a different diagnosis, like an eating disorder. So, all the focus gets put on that rather than maybe the underlying thing that could very well be BPD…”(GP, FG1)

Interestingly, GPs discussed how their suspicions that somatisation (the manifestation of psychological distress through the presentation of bodily symptoms) made their assessment and treatment more complicated. While these perceptions may be consistent with the roles of somatisation and somatoform disorders observed in presentations involving BPD, the obligation to avoid *assuming* a somatic component was nevertheless emphasised as a basic tenet of judicious primary care [11].
“… the majority of the physical symptoms are psychosomatic, but you can’t assume that. You have to look into it… So, every consultation is a long consultation.”(GP, FG1)

The presence of mental health comorbidities and the overlap in symptomatology, such as that seen between BPD and Bipolar Disorder contributed to difficulties in diagnosis. There was a “reluctance” to label the individual with personality disorder to *avoid adding to the* list of mental health issues.
“There’s a lot that I think are given the label bipolar disorder and complex post-traumatic stress disorder, and for some of my patients, there’s been a reluctance to actually use the label personality disorder.”(GP, FG1)

### 3.3. Difficulties with Patients’ Behaviour and the GP–Patient Relationship

GPs often elaborated challenges arising from complex presentations involving comorbidities, as well as diagnosis and related management issues, as being compounded by a set of difficult behaviours associated with patients with BPD. Participants expressed the view that these patients are particularly “troubled individuals”, and this seemed to undermine the GP–patient relationship when such patients presented repeatedly expressing high distress, anxiety, need, or anger. GPs described being overwhelmed and exhausted by the challenging healthcare requirements and nature of the relationship—so much so that one GP said they wanted to avoid treating BPD patients. Another GP articulated a sense of apprehension when describing what it is to have individuals with BPD in their practice.
[When describing their work with patients with BPD] “Sometimes I’ve got a patient that I think is a bottomless pool of need…”(GP, FG1)

One GP described a BPD patient as a “heart sink patient” to indicate how challenging such patients are to them. The GP acknowledged that it may also be experienced by the patient with respect to particular healthcare professionals. The GP described this realisation partly as a reminder for themselves to keep a *check on* such negative feelings that they may knowingly or unknowingly experience when encountering patients with BPD.
[When discussing the idea of “heart sink patients”] “I’ve got to balance that… now I wonder if there such a thing as a ‘heart sink doctor’. Could there be doctors that patients think, ‘oh, no, I’m not going to see that doctor’.”(GP, FG1)

GPs often mentioned that one of the prominent reasons for their reluctance to take on patients with BPD and continue to see them was that these patients frequently cancel appointments, often without sufficient warning, or that they fail to attend altogether. GPs plainly identified this issue as very frustrating, because it affects their ability to run their practices efficiently. They linked this issue to consultation duration and remuneration, which are factors driven by healthcare policy decisions that are part of a greater structural problem that is largely out of GPs’ control.
[When describing their reluctance to take on patients with BPD] “… I think a lot of GPs in their practice, they don’t refuse to see these patients… The problem is that they miss appointments frequently… A couple of patients of mine would easily miss 50% of their appointments, and a lot of doctors will just say, ‘no, two missed appointments, that’s it. I’m never going to see them again’.”(GP, FG1)

Participants discussed the lack of consistent attendance by patients with BPD alongside a recognition of their frequently changing living accommodations, which prompts their highly mobile nature as a patient group, and leads to obvious adverse consequences for engagement and continuity of care.
“… in the practice that I work in, a lot of the patients come and go… one of our problems is the patient group tend[s] to be a bit mobile. Sometimes they go down to the university clinic and sometimes they come to us and sometimes they go abroad… one of our problems is that we don’t get to know a lot of our patients really well.”(GP, FG2)

GPs also alluded to the intuitions or instincts that are often involved, especially when experienced GPs are attempting to make sense of a presentation involving a BPD diagnosis. GPs claimed that the experience of feeling “uncomfortable” was diagnostically useful, but it is also a factor that negatively impacts the interaction during consultation.
“I think it’s the boundaries, I think, when you feel like someone’s encroaching on your boundaries, your alarm bells go off… someone that makes you uncomfortable.”(GP, FG1)

Awareness of the stigma linked to BPD and the likelihood of management difficulties did not overrule the obligation to provide care. GPs highlighted their intention to provide quality healthcare in spite of the interpersonal and other difficulties associated with the diagnosis. One GP indicated that “the personhood of a patient must be somehow maintained”.
“People are people, they’re not diagnoses. So, you’ll keep on seeing them… and you know that, yep, she’s going to be needy and you hope that one day she’ll say, ‘yeah, today’s a good day’.”(GP, FG1)

While GPs clearly said that their attempts to be accommodating were important goals to pursue, GPs recognised that through such attempts, they may actually be *enabling* or *exacerbating* problematic behaviours among this patient group.
“So, it’s interesting because when you can see the clients and you sort of have more flexibility, then it can pose some different issues where you have to kind of be very mindful they don’t just come drop in all the time expecting you to sort of be there and be at their beck and call.”(GP, FG1)

### 3.4. Finding and Navigating Systems for Support

GPs spoke at length about the lack of support from and access to more specialist mental health advice when treating patients with BPD. Participants explained that they were an important gateway and connector for referral to further support services for these patients; however, they reported experiencing many challenges, for example, with finding specialist psychiatrists, constantly changing community services with short-lived funding, and time spent on the phone trying to find directories of services. GPs also discussed their appreciation of the collaboration, when it did occur, with psychiatrists and psychologists. These statements indicated that their difficulties regarding diagnosis, management, and problematic patient behaviours were at least partly alleviated by specialist input. Their frequent impressions that nothing they do seemed to be working was at least partially countered by collaboration, and they expressed a feeling of being reassured and supported with specialist contact.
“In my counselling, I very much take a person at face value and listen and try to understand their reality… why are they behaving like this and responding to their reality like this. Why do they not have more emotional resilience? I’m not very good at taking that step back, which is why I love getting letters from psychologists and psychiatrists…”(GP, FG1)

However, the reassurance and support that was received from specialist input was limited. They frequently mentioned systemic shortcomings concerning referral pathways, waiting times, and funding issues that impact BPD-specific treatment programs. Unfortunately, GPs saw navigating specialist referrals and arranging access to specialised programs as especially challenging, often leaving them to diagnose and manage patients with BPD on their own.
“… My challenge in General Practice is that psychiatrists are one of the most tricky people to access, and I think that is one of the challenges that will need to be overcome… you need to have a sense that there is a pathway or there’s somewhere to go…”(GP, FG1)

## 4. Discussion

Our observations and impressions of the data on which we based our analysis, and the consequent findings drawn from that analysis, were that the presence of service users (people with lived experience of BPD: patients and family carers), in particular, shaped the discussions positively, and facilitated GPs’ honest and open disclosures of the challenges and difficulties that they encountered with this patient group. The lived experience members of the research team were well-known local and national BPD advocates. GPs’ informal feedback at the conclusion of the focus groups was centred on how much they valued having people with lived experience ‘in the room’. 

### 4.1. Challenges Surrounding Diagnosis of BPD

Overall, GPs expressed varying feelings of uncertainty regarding their capabilities to diagnose BPD, which were accompanied by varying concerns about having no clear, evidence-based management routes to pursue following diagnosis. BPD is often undiagnosed in the primary care setting [2,4,23]. Our research suggests this might be partly a consequence of the clinical complexity and high rates of comorbidity that GPs encounter during presentations. GPs see such patients as demanding and emotionally exhausting, and need skills to be able to deal with these issues and feelings. That these encounters also typically occur during short consultation times further impedes GPs’ efforts towards careful diagnosis and effective management. GPs also often lack the training required to carry out formal psychotherapy that enables more effective diagnosis and ongoing management; however, they are able to use a range of other evidence-based strategies with some training [4]. 

Accurate diagnosis is vital for selecting appropriate management plans to address the BPD diagnosis and comorbid diagnoses. GPs’ practice may be informed by the view that BPD predisposes patients to developing other mental health disorders and that the management of BPD yields improvements in other disorders [2,4,11,24]. Importantly, unrecognised personality disorders may account for challenging GP–patient relationships; therefore, improving GP awareness that a personality disorder such as BPD is at play in the clinical picture may mitigate these interpersonal difficulties. Some of the clues and patterns that may prompt GPs to consider the presence of BPD are displayed in Table 2.

Since BPD symptomatology typically emerges in adolescence and early adulthood, GP assessment may benefit from long-term observation in patients that have been part of a GP practice over a long duration of time since childhood; crucially, this also affords the opportunity to act early and initiate prevention/early intervention measures [4,23,27]. Prevention/early intervention is also important for mitigating the development of frequently co-occurring disorders, because BPD is recognised as a risk factor for other mental health disorders [2,4,10,11]. However, as described by many of the GPs during the focus group discussions, the attendance issues and highly mobile nature of this patient group can prevent long-term observation and lead to a discontinuity of care.

Overlap in symptomatology, compounded by high rates of comorbidity among BPD and other mental health disorders, including other personality disorders, can lead to GPs failing to make the diagnosis of BPD altogether [4,10,26,28]. Despite sharing clinical features with several other mental health disorders, BPD exists as a distinct psychiatric entity that can be diagnosed by experienced healthcare professionals [4,5,26,28,29]. Comprehensive and careful diagnosis of BPD is central: it is the first step towards the effective management of BPD itself and the mental health disorders that may be co-occurring [2,10,11]. The NHMRC Guideline emphasises this view, acknowledging that the management of BPD is very different from the management of other mental health disorders in patients who do not have BPD [4]. Hence, GPs should make judicious diagnosis a major goal, which may necessitate a second opinion, specialist input, and referral to mental health assessment teams [4].

The confusion regarding the diagnosis of BPD in the presence of overlapping symptoms can be exemplified by considering BPD in conjunction with other mental illnesses, such as Major Depressive Disorder, Bipolar Disorder, and other personality disorders. For example, up to 20% of patients diagnosed with BPD also have comorbid Bipolar Disorder, and approximately 15% of patients diagnosed with Bipolar Disorder have comorbid BPD [30]. If a diagnosis of Bipolar Disorder has been made in a patient who actually has BPD, this can lead to unrealistic management expectations regarding the effects of medications, and it can prevent the healthcare practitioner from pursuing effective psychosocial strategies, such as those based on dialectical–behavioural therapy [25,26,28]. On the other hand, if someone receives a diagnosis of BPD instead of Bipolar Disorder, they may be deprived of potentially effective pharmacotherapy (such as mood stabilisers) that leads to significant improvements in the course of Bipolar Disorder [25]. Making these distinctions is crucial, because they help lead to the proper diagnosis of BPD and of co-occurring mental health disorders, allowing management plans to be initiated that have been widely acknowledged and recommended in clinical guidelines [2,4,10,11,24]. 

The focus group discussions highlighted that GPs wanted to learn more about BPD and improve their skills and response to patients with BPD. However, there is a wide spectrum of GP confidence and skills to accurately diagnose and manage BPD, especially in the setting of great clinical complexity involving comorbidity. Participants also noted that the referral process can be complicated, waiting times for specialists are long, and that mental health programs often suffer from funding issues. The feeling was that, in many cases, GPs are unsupported, and that a comprehensive evidence-based guideline would be very helpful. 

Interestingly, the NHMRC Guideline was released three years before the focus group discussions, which may point to a failure in adequate awareness, dissemination, and implementation of the document. However, there are two further problems with the current guideline: the document is 182 pages long; and only nine of the 64 recommendations have a good evidence base; most of the remaining recommendations are either consensus or practice points. A two-page summary would be more user-friendly to GPs, and also more likely to be read by them. Whilst the guideline stresses that effective implementation at the local level requires an assessment of the relevant barriers and enablers to best practice with a view to optimising clinician uptake [4], there must be further efforts to improve these processes.

### 4.2. Comorbidities and Clinical Complexity

The focus group results reveal that one of the greatest challenges that was recognised in managing patients with BPD is that their presentations are often highly complex and involve high rates of comorbidity, and that this situation has not changed, since these issues were first raised in the literature almost two decades ago [2,5,10,29,31,32]. 

Health system boundaries constrain treatment and management; this was also expressed by the participants. Comorbidity and clinical complexity are made especially problematic by the time constraints that are typical of consultations in primary care. This is consistent with studies that reveal the increasing demands put on GPs to manage more patients in shorter amounts of time due to the increasing burden of chronic disease [12,33]. In response to an apparent need for systemic and structural primary healthcare changes, the World Health Organisation has promoted raising awareness among policy-makers and healthcare providers that presentations involving *comorbidities* are to be expected among individuals with chronic conditions [33].

#### 4.2.1. Mental Health Comorbidities

As stated above, BPD is known to co-occur with several other mental health problems, and is most strongly associated with mood disorders, anxiety disorders, and substance use disorders [3,5,26,29,31]. There is also a significant relationship between BPD and eating disorders and trauma and stressor-related disorders (such as post-traumatic stress disorder, or PTSD) [31,32]. When several mental health disorders co-exist during presentation, alongside difficulties establishing diagnoses, GPs may face uncertainty about appropriate management, including the role of medications [3,4,25,26,28]. The GP participants in the current study clearly described their struggles with managing treatment options for patients with BPD. Even in cases where a BPD diagnosis is made, it is often unclear to GPs how they ought to proceed with the management of several co-occurring mental health disorders [10,26,28]. However, it is thought that patients with BPD require management that puts the primary emphasis on the BPD diagnosis, regardless of which other mental health comorbidities are present [4,10,11]. Major depression, for example, may remit (and stay in remission) upon the successful management of BPD [1,3]. Conversely, Gunderson found that a reduction in major depression symptoms through targeted management *did not* lead to a significant reduction of BPD symptoms [3]. Targeting BPD and implementing a psychotherapy-based treatment plan has been shown to improve the courses of BPD and other comorbid mental health disorders [10,11]. GPs in the current study expressed a strong desire to ‘care’ for their patients with BPD, akin to providing a more psychotherapeutic, evidence-based response. They wanted to feel assured that they were ‘doing the right thing’ for these patients; however, they also felt that they needed more knowledge and skills in this area. 

#### 4.2.2. Medical Comorbidities

Studies investigating BPD in the primary care setting have also shown that BPD co-occurs with a range of medical disorders, including cardiovascular disease, obesity, diabetes, back pain, chronic pain, urinary incontinence, hepatic and other gastrointestinal disease, fibromyalgia, migraine, chronic fatigue, arthritis, and sexually transmitted infections [2,3,29,34,35,36]. Similar to our findings, other studies have revealed that GPs frequently address multiple medical complaints among individuals with BPD, and that the presence of medical comorbidities plays a significant role in the clinical difficulties faced by GPs during presentation [13,37,38]. They imply that a vicious cycle exists wherein the presence of chronic medical disorders may hinder recovery from BPD, and that the presence of BPD symptomatology may worsen the course of medical disorders [13]. These possibilities may very well have practical applications for GP management strategies.

Keuroghlian et al. [13] explored the longitudinal interaction between a diagnosis of BPD and the course of chronic medical illness, health-related lifestyle choices, and healthcare service utilisation in a 10-year prospective follow-up study. One of their key findings was that patients who did not recover from BPD suffered from more chronic medical disorders over a 10-year period compared to patients who did recover from BPD over the same period of time [13]. Such findings point to a close connection between BPD and poor physical/medical health status; a notable implication is that the successful remission of BPD may be important for managing medical comorbidities, just as it is for managing mental health comorbidities. 

Using the idea of a vicious cycle may help explain and alleviate the challenges of tackling the medical aspect of presentations involving BPD. Establishing clear guidance for GPs that is acceptable and feasible with their practice would be important to ensure that acknowledging this complexity does not turn GPs away from providing such care to these patients. The management of medical comorbidities may be impeded by the increased rates of problematic and unhealthy activities that are linked to BPD psychopathology, such as cigarette smoking, excessive alcohol use, prescription medication overuse, lack of regular exercise, and risk-taking behaviour [2,13]. It seems reasonable then that in presentations involving a BPD diagnosis, GP plans should target maladaptive behaviours that are characteristic of BPD in order to achieve medical treatment goals, rather than treating such behaviours separately and distinctly from the wider picture of patient healthcare needs [4]. This approach, of course, requires the ability to establish diagnosis and familiarity with relevant management strategies.

Furthermore, managing medical complaints is complicated by the phenomenon of somatisation, which was an observation that featured during the discussions. The recognition of somatisation may be a vital element of the GPs’ skill set that could go a long way in mitigating the challenges they face during presentations involving BPD. Studies have revealed that individuals who experience somatisation are highly represented among patients utilising primary care, and that there is a significant relationship between somatisation and BPD [37,39]. These findings further suggest that GPs may benefit from establishing a BPD diagnosis, understanding relevant management, and taking note of how somatisation figures into presentations. Taking the role of somatisation into consideration may reduce the overwhelming complexity arising from multiple presenting medical complaints. 

### 4.3. Difficulties with Patient Behaviour and the GP–Patient Relationship

The results corroborate the significance of behavioural difficulties among patients with BPD and the challenging nature of the doctor–patient relationship, which can affect clinical practice [2,3,4,11]. GPs agreed that their relationships with these patients are laden with interpersonal difficulties that interfere with management goals. GPs also pointed out their frustrations with other problematic behaviours, such as frequent cancellations, non-attendance, and crisis presentations, which feed into the negative GP–patient relationship. These frustrations have been documented in other studies [2,6]. Nevertheless, during the focus group discussions, GPs clearly demonstrated their commitment to optimal care, and that they were trying their best, in spite of these challenges. 

Many of the frustrations and uncomfortable relationships that GPs experience can be couched in terms of transference and countertransference. These clinical phenomena are known to be prevalent during interactions with individuals with BPD, particularly during episodes of affective dysregulation by patients, which might compromise the GP–patient relationship [2,3,4,11]. Our sense from hearing these GPs recount their experiences and frustrations was that they felt significant pressure from many of their patients with BPD, which was exacerbated by their own perceived limited skills to know how to help these patients. Therefore, GPs were put in a position of continually needing to navigate the professional boundary with the patient with BPD, and they did not like how this made them feel; that is, they felt disempowered and frustrated, given they had a genuine desire to help, but also felt overwhelmed with the needs and emotions that such patients expressed to them.

It may also be that transference /countertransference issues are a major factor in GPs becoming more or less active in assisting such patients. Our results revealed that whilst GPs were very frustrated in their attempts to provide care to patients with BPD, and were aware of issues of transference and countertransference, they also perceived few supports available to them from more specialist mental health service providers to improve their circumstances, which is similar to the findings of others [39]. Without such support, GPs were left with feelings of apprehension, dismay, and even disempowerment, as evidenced by their description of these as “heart sink” patients. This phenomenon was pervasive in a similar study [14]. Recognising that BPD is present in the first place can help GPs respond and act appropriately to challenging relationships with BPD patients [6]; however, responsibility cannot rest only with the GP. Support from and coordination with other parts of the healthcare system are needed. Case discussion training that involves local networks of GPs have been proposed as one solution to GPs’ needs [40]. Interestingly, overwhelming feedback from GP participants about the design of the current study was that it provided them with an opportunity to share, vent, reflect, and learn more about the management of BPD.

Patients’ underlying fears of abandonment, splitting, and devaluation–idealization play a large role in the GP–BPD patient relationship. Such fears can be triggered by a GP being unavailable to the patient such as when a patient requests a last-minute appointment, unscheduled medication refill, or after-hour phone call. GPs identified these problems during the focus group discussions, indicating that they often are aware that these phenomena are at play, but they did not express confidence in dealing with these problems [2,3]. The structure of general practice also did not always lend itself to being responsive to the needs of BPD patients making such requests. GPs struggle with adopting patient management strategies that do not have other adverse repercussions. They mentioned that having more flexibility towards patients with BPD can pose problems such as instilling the expectation that patients can “drop in all the time” and that GPs should be at their “beck and call”, which is consistent with the idea that patient behaviours and GP responses perpetuate GP–patient relationship problems. Demonstrating too much flexibility in an attempt to improve management may have negative consequences for the relationship and management goals. 

Clinical practice guidelines and most studies investigating BPD management recognise that there are several problems stemming from patient behaviours and the GP–patient relationship [2,3,4]. These sources offer practical solutions that GPs can incorporate into management plans, and in fact, most clinical advice emphasises the importance of mitigating problematic behaviours and carefully controlling the dynamic of the GP–patient relationship in order to achieve remission [2,3,4] (see Table 3). These solutions address most of the problems that GPs mentioned in the focus group discussions relating to patient behaviour and the GP–patient relationship.

### 4.4. Systems Issues

Based on the findings, several system-level improvements are suggested to improve GPs’ role in supporting patients with BPD. These are summarised in Table 4. 

### 4.5. Limitations

The study was conducted in the evening and at one central location, with the aim of enhancing GPs’ participation. Despite this, the study involved a small sample. More detailed individual demographic information about GP participants (such as age and years of experience, location, and structure of their clinic) were not collected. These differences could have added further interesting information to understanding GPs’ experiences and the data analysis. They were also a purposive, self-selected sample of GPs. Therefore, the findings may not reflect the experiences of the broader GP population in Australia, GPs in rural and remote locations, the needs of GPs in other countries, or GPs working within different healthcare practice and funding structures. Further research with a larger sample and more rigorous methods is needed in order to examine the connection between attitudes towards providing care to patients with BPD and the health service context for GPs.

Those with a greater interest in providing primary care for patients with BPD may have been more likely to attend the focus groups. This is also a strength of the findings in highlighting the depth of issues that GPs experience, especially among those with the desire to share their experience. If those with an interest in patients with BPD report significant problems in supporting these patients, then it is likely that those who are less engaged with this patient group are likely to experience even greater concerns, as well as skills and support gaps. Whilst the study set out to also explore GPs’ engagement with carers of patients with BPD, little data on this aspect was recorded because, during the focus group discussions, GPs were predominantly focussed on discussing the many challenges that they faced in their practice with BPD patients. Given the important role that family can provide to patients, this is an area needing further investigation within the primary care context. The co-design of the study through a collaboration between researchers, clinicians, and people with lived experience was seen as a strength of this study.

## 5. Conclusions

The current study highlighted many challenges faced by GPs in their struggles to provide care to patients with BPD. This included issues regarding the diagnosis of BPD alongside mental and physical health comorbidity, and the need for more evidence-based management strategies to respond effectively to the complexities that are apparent for these patients in the primary care setting. BPD has been reported as the most psychologically challenging condition for GPs to treat [3]. The primary care setting is an important point of contact and support for patients with BPD, and GPs are ideally placed to provide access to care for these patients in their community. This study highlighted various clinical and systemic barriers that appear to require further education and support to GPs, as well as healthcare policy changes, to enable a greater knowledge translation of effective strategies and address service gaps. Health service pathways for this high-risk and high-need patient group are dependent on the quality of care that GPs provide, which is in-turn dependent on GPs’ capacity to identify and understand BPD, and be supported sufficiently to develop key skills that are necessary to provide effective support and treatment for patients with BPD [3]. Healthcare policy, to the extent that it dictates the temporal organisation of primary care, clearly emerged as a prominent structural factor obstructing GPs’ attempts to properly address multiple comorbidities for patients with BPD. Several strategies are suggested to support GPs’ role in supporting patients with BPD. 

## Figures and Tables

**Figure 1 ijerph-15-02763-f001:**
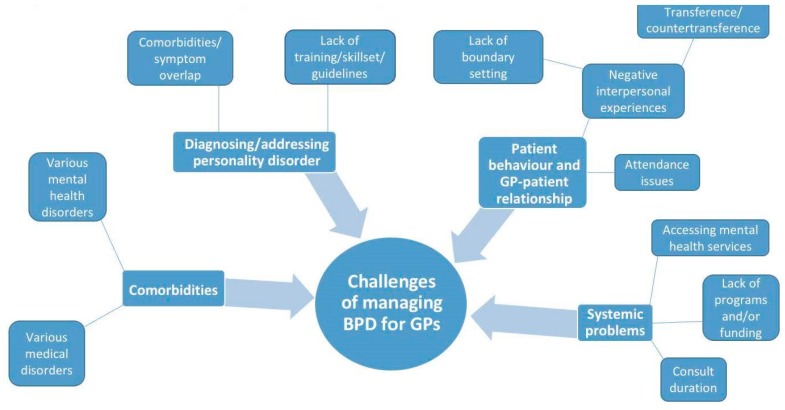
Summary of Findings: mind map.

**Table 1 ijerph-15-02763-t001:** Focus Group Guide Questions. BPD: Borderline Personality Disorder.

Degree of interaction with patients with BPD What is the prevalence of BPD in your medical community of practice? What is the number of BPD patients in your practice? Views about treating patients with BPD Are there rewarding experiences when working with people with BPD? Diagnostic process What guides your assessment of BPD? Are there any issues in making (and delivering) the diagnosis of BPD? Challenges in working with patients with BPD Can you tell us about your confidence to work with this population? Is there anything you would like to improve/change when working with these patients? Can you tell us about how you deal with self-harm and other crisis in these patients? Treatment pathways, services, and resources available to support people with BPD in primary care To which services do you refer BPD patients on a regular basis? What training needs do you have to support your work with these patients? We are interested in your views of effective and less effective services. Carers and social context for the person with BPD How have you worked with carers? Any challenges in engaging with carers? What has worked well? How well do you think you have been able to provide advice and guidance to carers about caring for people with BPD in different states of need? Possible areas for improvement in primary care responses to BPD

**Table 2 ijerph-15-02763-t002:** Clues to Making a BPD Diagnosis.

The patient: Presents during a crisis Showing intense emotional distress Reports of suicidal ideation Reports of previous suicide attempts Shows signs of recurrent self-harm (e.g., cut marks) Reports of risk-taking behaviour Reports of polypharmacy Makes inappropriate medication requests Reports of various relationship problems that seem to be long-term or recurrent [2,4,11,12,23]. An Example: Is it BPD or Bipolar Disorder? BPD and Bipolar Disorder can be partly distinguished by exploring the time course related to mood lability: The mood shifts in BPD are not sustained, and patients may report mood shifting over a span of minutes or hours, while the mood shifts in Bipolar Disorder are typically sustained over longer durations of time [25,26]. The mood shifts in BPD are often preceded or aroused by incidents involving interpersonal sensitivity, whereas the mood shifts in Bipolar Disorder are typically autonomous or without obvious stressors [25,26]. NB. In cases where history-taking is being carried out for assessment of BPD, the process itself can be a distressing experience that re-traumatises some individuals [4].

**Table 3 ijerph-15-02763-t003:** Strategies for Managing the GP–BPD Patient Relationship. GP: general practitioner.

**Set boundaries for the relationship with BPD patients**: This is a skill that needs to be taught and may take some practice, for both the GP and patient with BPD [4]. **Use “management contracting**”: a negotiation between GP and patient about the expectations during treatment, and also makes clear that responsibility for management is *shared* by both parties; this encourages accountability and independence in the patient [3]. **Discuss and identify short-term and long-term goals with the patient,** encouraging them to reduce inpatient admissions and reliance on health services, as well as practicing self-regulation in their non-clinical interpersonal encounters [4]. **Schedule regular structured appointments** for check-ins and avoiding only crisis presentations [2,3,4], and addressing perceived abandonment [2]. **Develop a “crisis management plan”:** agreed upon by the GP, patient, and other care providers; this may reduce excessive reliance on GP services and facilitate self-management skills [3,4]. “Crisis plans” also aim to minimise emergency department admissions, which occur frequently among individuals with BPD [3,4]. **De-escalate any possible emotionally intense confrontations with a calm and neutral demeanour** [4]. **Ensure that frequent communication is maintained between various care providers**: important for preventing patients from “playing one practitioner against another” through the process of splitting and devaluation–idealization [3]. **Management goals should be realistic and take into account the fluctuating and long-term course of BPD symptomatology:** GPs must bear in mind that although significant improvements and remission are possible, they occur over the long-term [28]. While boundary-setting and management contracting are emphasised, **management plans should also allow a degree of flexibility to accommodate changing circumstances** [4].

**Table 4 ijerph-15-02763-t004:** Improvements Needed to Support GP Practice with Patients with BPD.

The NHMRC guidelines need to be updated and streamlined in order to make them more useful and accessible to GPs. Assistance to GPs, through education and consultation liaison with mental health specialist support, to make a clear mental health diagnosis for those patients with BPD. Specific education and post-education support to GPs, to enable them to translate knowledge acquired to their practice with patients with BPD. Ensure that guidelines developed for various conditions consider addressing co-morbid mental health conditions such as BPD. Improved communication systems to ensure easier access by GPs to mental health specialists and BPD-specific treatment programs. Greater collaboration between State and Commonwealth-funded services to connect GPs better to support resources for BPD, such as a central electronic resource by area. Explore alternative payment schemes for GPs who manage patients with complex health care needs such as BPD.

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
