# Peer review of "Exploring General Practitioners’ Views and Experiences of Providing Care to People with Borderline Personality Disorder in Primary Care: A Qualitative Study in Australia"

_ijerph, 2018, doi:10.3390/ijerph15122763_

Round 1

Reviewer 1 Report

This paper explores GP’s experiences of providing care to individuals with BPD. This is an important topic as we need to better understand how practitioners perceive and experience service users with these traits to improve practice. I enclose some specific comments below.

 Intro

Line 73 – word missing- “BPD to assist them navigate existing services”

Line 85 – The paragraph beginning on line 85 would be improved if it clarified also that not all experiences with GPs are negative. Patients’ also describe positive aspects of their care, and it is important to acknowledge this (e.g., see Veysey, 2014). At present, the focus inadvertently paints a slightly negative picture of GPs.

Veysey, S. (2014). People with a borderline personality disorder diagnosis describe discriminatory experiences. Kōtuitui: New Zealand Journal of Social Sciences Online, 9, 1.

Methods

Line 137 – word missing “This visual process was chosen because enabled”

The inclusion of academics and service users alongside the 12 GPs suggests a total N of 22, but these data are not reported. Rather, only the GP data are reported. The authors rationalise their recruitment strategy and approach to the focus group, although I am not convinced that a “stakeholder” group was deliberate and I wonder whether this study was an afterthought. The fact that no demographic data were collected reinforces this view. In spite of this, the study adds value and I would welcome its publication.

The total n of 12 and not 22 needs clarifying in the Abstract and Method section. Moreover, whilst this method has benefits, it has limitations that need acknowledging and discussing within the Discussion section. For example, to what extend did the presence of service users shape the discussions and hamper GPs’ honest and open disclosures of the challenges/difficulties they encounter with this client group?

Results

It isn’t clear why demographic information was not collected. There is no solution to this other than to discuss the potential problems with this as has been done.

Line 173: should “is” say “if”?: “what is somebody abandons me”

The results were well reported and made for an interesting read.

Discussion

Line 415 – “The results are consistent with findings in the literature that affirm BPD is often undiagnosed in the primary care setting”.  I don’t think the authors’ can convincingly argue this given the small sample. The findings of this study highlight the challenges in diagnosing and in some cases this ‘could’ have led to cases not being diagnosed, but whether this often happened is not clear from this evidence. I suggest rewording.

Section 4.2.1 would benefit from application/explicit reference to the present findings.

For the most part this is an incisive discussion and the clinical implications are clearly stated.

Conclusion

Remove line “This section is mandatory. Please summarize the main achievements and/or results in 624 this section”

The conclusions section could be strengthened by removing the first paragraph and perhaps shifting one or two arguments from this to the end of the second paragraph so that it ends by concluding with the main clinical implication of the study.

Author Response

1.     Intro

Line 73 – word missing- “BPD to assist them navigate existing services”

Response: We have corrected this error.

2.     Line 85 – The paragraph beginning on line 85 would be improved if it clarified also that not all experiences with GPs are negative. Patients’ also describe positive aspects of their care, and it is important to acknowledge this (e.g., see Veysey, 2014). At present, the focus inadvertently paints a slightly negative picture of GPs.

Veysey, S. (2014). People with a borderline personality disorder diagnosis describe discriminatory experiences. Kōtuitui: New Zealand Journal of Social Sciences Online, 9, 1.

Response: We have added a sentence to reflect this sentiment, and also reviewed and cited Veysey (2014) to support this idea.

3.     Methods

Line 137 – word missing “This visual process was chosen because enabled”

Response: We have corrected this error.

4.      The inclusion of academics and service users alongside the 12 GPs suggests a total N of 22, but these data are not reported. Rather, only the GP data are reported.

Response: We have added a statement at the end of the methods section about only reporting GP data. We have also added a comment to the limitations section.

5.      The authors rationalise their recruitment strategy and approach to the focus group, although I am not convinced that a “stakeholder” group was deliberate and I wonder whether this study was an afterthought. The fact that no demographic data were collected reinforces this view. In spite of this, the study adds value and I would welcome its publication.

Response: We thank the reviewer for raising this concern. In fact, the stakeholder group was deliberate. They were a dedicated ‘research’ team that met regularly over several weeks to plan this research activity. It was very much a shared and planned effort between dedicated academics, clinicians and people with lived experience of BPD. We have added more information about how the team planned the design of the study.

6.      The total n of 12 and not 22 needs clarifying in the Abstract and Method section.

Moreover, whilst this method has benefits, it has limitations that need acknowledging and discussing within the Discussion section. For example, to what extend did the presence of service users shape the discussions and hamper GPs’ honest and open disclosures of the challenges/difficulties they encounter with this client group?

Response: We have added further clarification in the abstract and methods sections, as requested. We have also added some dialogue about the impact of this at the beginning of the discussion section, as suggested. Your question is an interesting one. Our sense is that it was an extremely honest and open disclosure and that the presence of people with lived experience helped to facilitate this. The consumers and carers who were present are ‘seasoned’ and well known in the BPD space in South Australia; some are nationally recognised BPD and mental health advocates. We have made further comment in the limitations section about the strength of this co-designed research.

7.     Results

It isn’t clear why demographic information was not collected. There is no solution to this other than to discuss the potential problems with this as has been done.

Response: We did collect very basic demographic information about the GP participants but did not report it initially because it was not as comprehensive as we would have done in other qualitative studies. We have now provided the demographic information that was available to us. We have already acknowledged this as a limitation.

8.      Line 173: should “is” say “if”?: “what is somebody abandons me”

Response: We have corrected this error.

9.      The results were well reported and made for an interesting read.

Response: thank you.

10.  Discussion

Line 415 – “The results are consistent with findings in the literature that affirm BPD is often undiagnosed in the primary care setting”.  I don’t think the authors’ can convincingly argue this given the small sample. The findings of this study highlight the challenges in diagnosing and in some cases this ‘could’ have led to cases not being diagnosed, but whether this often happened is not clear from this evidence. I suggest rewording.

Response: We have revised this sentence by removing the first section of the sentence, given that the sentence following it provides clarification in relation to what our study contributes an increased understanding of why this might be so.

11.  Section 4.2.1 would benefit from application/explicit reference to the present findings.

Response: Thank you for picking up this omission. We have added wording in two places within this section, to apply these ideas back to our findings.

12.  For the most part this is an incisive discussion and the clinical implications are clearly stated.

Response: thank you

13.  Conclusion

Remove line “This section is mandatory. Please summarize the main achievements and/or results in 624 this section”

Response: Thank you. We apologise for this oversight, and have removed this text.

14.  The conclusions section could be strengthened by removing the first paragraph and perhaps shifting one or two arguments from this to the end of the second paragraph so that it ends by concluding with the main clinical implication of the study.

Response:

Reviewer 2 Report

This qualitative study utilized focus groups of general practitioners, people with borderline personality disorder (BPD), family care providers and academics and thematic analysis to capture the experience of providing primary care to people with BPD. The study results are well presented and contribute to our understanding of the challenges in caring for patients with BPD. Two issues should be raised in the discussion that might put the current findings into a broader context. Discussion of the difficulties with patient behaviour and the GP-patient relationship did not raise the possibility of the relationship leading to boundary violations. As suggested physicians at risk for boundary violations are unlikely to volunteer for such focus groups. The lack of information about the setting in which these family physicians practice does not clarify the barriers that exist to accessing mental health care. In many settings in North America, family physicians may have access to integrated care with mental health professionals and problems of access should greatly be diminished.

Author Response

1.     This qualitative study utilized focus groups of general practitioners, people with borderline personality disorder (BPD), family care providers and academics and thematic analysis to capture the experience of providing primary care to people with BPD. The study results are well presented and contribute to our understanding of the challenges in caring for patients with BPD. Two issues should be raised in the discussion that might put the current findings into a broader context.

Response: We thank the review very much for their time and support of our work.

2.     Discussion of the difficulties with patient behaviour and the GP-patient relationship did not raise the possibility of the relationship leading to boundary violations. As suggested physicians at risk for boundary violations are unlikely to volunteer for such focus groups.

Response: We thank the reviewer for this insightful comment and have added further wording to the discussion in section 4.3 to clarify the ideas already there, adding specific reference to the boundary issue. Our sense from hearing these GPs recount their experiences and frustrations was that they felt significant pressure from many of their patients with BPD which was exacerbated by their own perceived limited skills to know how to help these patients, and the lack of support they perceived from mental health services. “They were therefore put in a position of continually needing to navigate the professional boundary with the patient with BPD and they didn’t like how this made them feel; that is, they felt disempowered and frustrated, given a genuine desire to help, but also feeling overwhelmed with the needs and emotions that such patients expressed to them.”

3.     The lack of information about the setting in which these family physicians practice does not clarify the barriers that exist to accessing mental health care. In many settings in North America, family physicians may have access to integrated care with mental health professionals and problems of access should greatly be diminished.

Response: We have added some information about the Australian setting in which this study was set. This appears near the beginning of the Materials and Methods section (p.3). Unfortunately, in Australia, general practice settings and mental health services are quite distinct service types in which communication and referral pathways can be quite fragmented and difficult to navigate.